# Study on Dispersion, Adsorption, and Hydration Effects of Polycarboxylate Superplasticizers with Different Side Chain Structures in Reference Cement and Belite Cement

**DOI:** 10.3390/ma16114168

**Published:** 2023-06-02

**Authors:** Yunhui Fang, Xiaofang Zhang, Dongming Yan, Zhijun Lin, Xiuxing Ma, Junying Lai, Yi Liu, Yuliang Ke, Zhanhua Chen, Zhaopeng Wang

**Affiliations:** 1Polytechnic Institute, Zhejiang University, Hangzhou 310015, China; fangyunhui@126.com; 2College of Civil Engineering and Architecture, Zhejiang University, Hangzhou 310058, China; dmyan@zju.edu.cn; 3KZJ New Materials Group Co., Ltd., Xiamen 361101, China; 15859120891@163.com (X.Z.); geniusvcap@163.com (Z.L.); mary@xmabr-kzj.com (X.M.); yuliang_ke8207@163.com (Y.K.); zhanhuachen123@126.com (Z.C.); wangzp1109@126.com (Z.W.); 4School of Materials Science and Engineering, Zhejiang University, Hangzhou 310023, China; liuyimse@zju.edu.cn

**Keywords:** polycarboxylate superplasticizer, cement, molecular structure, flowability, hydration

## Abstract

To investigate the effects of Reference cement (RC) and Belite cement (LC) systems, different molecular structures of polycarboxylate ether (PCE) were prepared through the free radical polymerization reaction and designated as PC-1 and PC-2. The PCE was characterized and tested using a particle charge detector, gel permeation chromatography, a rotational rheometer, a total organic carbon analyzer, and scanning electron microscopy. The results showed that compared to PC-2, PC-1 exhibited higher charge density and better molecular structure extension, with smaller side-chain molecular weight and molecular volume. PC-1 demonstrated enhanced adsorption capacity in cement, improved initial dispersibility of cement slurry, and a reduction in slurry yield stress of more than 27.8%. LC, with its higher C_2_S content and smaller specific surface area compared to RC, could decrease the formation of flocculated structures, resulting in a reduction in slurry yield stress of over 57.5% and displaying favorable fluidity in cement slurry. PC-1 had a greater retarding effect on the hydration induction period of cement compared to PC-2. RC, which had a higher C_3_S content, could adsorb more PCE, leading to a greater retarding effect on the hydration induction period compared to LC. LC and PC-2, on the other hand, exhibited inhibition during the hydration acceleration period. The addition of PCE with different structures did not significantly affect the morphology of hydration products in the later stage, which was consistent with the trend of K_D_ variation. This indicates that the analysis of hydration kinetics can better reflect the final hydration morphology.

## 1. Introduction

With the rapid development of China’s economy and the large-scale construction of civil engineering, the construction industry in China is moving towards high efficiency, high performance, and sustainability. The application of polycarboxylate superplasticizer (PCE) is becoming increasingly popular. PCE has carboxyl groups on the main chain structure, which can be adsorbed on the surface of cement particles, providing electrostatic repulsion, and the branching of the PCE can form steric hindrance, hindering the cement particles from approaching each other and releasing water from the flocculated structure between the cement particles, thereby achieving the effect of dispersion and water reduction [1], as shown in Figure 1. Andersen [2], Yamada [3], Saka [4], Plank [5], and others have conducted research on the relationship between the main chain and side chain structure of PCE and its performance. Liu Jiaping [6] and others believed that the spatial repulsion provided by the side chains of PCE was the main driving force for dispersion. The grafting density is high, resulting in a high coverage of the material surface, which increases the spatial repulsion between the cement particles, thus improving the dispersion efficiency. Kong Xiangming et al. [7,8] found that the higher the carboxyl group content in PCE, the more carboxylate functional groups on the main chain will provide negative charges. When adsorbed on the surface of cement particles in the cement slurry through charge interaction, electrostatic repulsion and steric hindrance provided by the side chains of PCE will break up the flocculated structure between cement particles, resulting in better dispersion of the cement slurry.

Rheological parameters are important indicators to characterize the workability and rheological properties of freshly mixed cement paste. The internal structure of the cement paste can be characterized by the rheological properties of the paste, and the changes in the internal structure can be analyzed by rheological parameter analysis. Frank Winnefeld et al. synthesized polycarboxylates using ester monomers with different degrees of polymerization and methacrylic acid, and the research results showed that the lower the side chain density, the better the initial dispersibility of cement slurry, the lower the yield stress, and the lower the plastic viscosity of the cement paste [9].

The adsorption of superplasticizers and cement hydration occurs simultaneously and is a dynamic process. However, the mechanism of interaction between PCE and cement, especially the relationship between PCE molecular structure and cement interaction, is still not unified. The adsorption capacity of PCE largely depends on its side-chain carboxyl density, which enhances its adsorption capacity on the positively charged surface of cement particles, increases the thickness of the adsorbed layer on cement particles, and improves the dispersibility of the cement paste [10,11]. Elzbieta, J.R. [12] believed that when the length of the main chain of polycarboxylate molecules is long, the molecular weight is large, it is easy to adsorb on the surface of cement particles, increasing the coverage area of a single molecule. Liu Jiaping et al. [13] found that the adsorption characteristics of PCE on the same mineral were similar, mainly adsorbing on the aluminate phase and its hydration products. PCE reached adsorption equilibrium with the silicate phase at lower dosages, and the saturation adsorption capacity was small. In the C_3_A system, the adsorption amount of PCE was linearly related to its dosage, and there was no adsorption saturation point in the dosage range of 0–8 mg/g. Zhao Yue et al. [14] found that PCE could intercalate into the hydrated calcium aluminate and form organic metal mineral phases, which gradually transformed into AFm in the cement hydration system. In addition, PCE promoted the formation of AFt and AFm during the early stage of cement hydration, which reduced the adsorption capacity after 1 h. The adsorption amount and adsorption form of PCE on cement particles have different effects on the charge of individual cement particles and adjacent cement particles. When the particles are close to each other, the adsorbed PCE due to electrostatic and steric effects causes the particles to repel and disperse, thereby releasing free water between the cement particles. The different structural forms of PCE and the interaction between the cement particles during the early stage of cement hydration result in different rates of mineral dissolution, pore solution concentration, hydration product nucleation rate, and growth rate.

When cement comes into contact with water, mineral hydration reactions occur, and the hydration heat-release curve can be divided into five stages, including (I) the initial period, (II) the induction period, (III) the acceleration period, (IV) the deceleration period, and (V) the stable period [15]. During the induction period, after water is added to the cement, the mineral phases gypsum and C_3_A dissolve rapidly, and AFt is formed. At the same time, the concentrations of Ca^2+^, OH^−^, and SO_4_^2−^ ions in the cement solution continue to increase. When the ion concentration reaches a certain value, the nucleation and crystallization processes occur at the solid–liquid interface, resulting in the formation of calcium hydroxide (CH) and calcium aluminate (AFt) crystals. Meanwhile, tricalcium silicate (C_3_S) is also gradually hydrated, forming a small amount of short, fibrous calcium silicate hydrate gel (C-S-H). During the induction period, gypsum further dissolves, and the concentrations of Ca^2+^ and SO_4_^2−^ ions continue to increase. Influenced by nucleation or diffusion-controlled reactions, a large amount of CH nucleation and C-S-H nucleation begins, and AFt stably exists in the solution [16]. During the acceleration period, the solution reaches the supersaturation required for the nucleation and growth of hydration products. Hydration products begin to nucleate and grow in large quantities, and the pore solution continues to provide the ions necessary for growth. As the hydration proceeds, the nucleation sites and growth rate continue to increase, and the acceleration period enters the highest heat-release rate. Subsequently, the pore solution continues to decrease while the hydration products come into close contact and the ion diffusion rate decreases, entering the deceleration period. In the deceleration period, the remaining pore solution and unbound water are further consumed, and AFt transforms into monosulfate AFm due to the exhaustion of sulfate ions. C_2_S also participates in the particle diffusion reaction and hydration between the particles, and the paste porosity continues to decrease and becomes denser [17,18].

Polycarboxylate superplasticizer adsorbs on the surface of cement minerals, preventing the dissolution of cement minerals and the nucleation of C-S-H, thus hindering cement hydration reactions [19]. The increase in carboxyl group content on the PCE main chain enhances the strong chelation between Ca^2+^ ions, thus delaying the induction period of hydration, but it has a promoting effect on the acceleration period of hydration [20,21]. There are still significant differences in the conclusions about the molecular structure of PCE and the inhibition of C-S-H nucleation and growth by PCE proposed by Zuo Yanfeng [22].

RC is composed of various clinker minerals, and the quality of cement varies among different manufacturers. The surface properties of cement particles, such as surface area, porosity, polarity, and the number of active sites, can affect the adsorption and hydration of water [23]. The adsorption of superplasticizers differs among the different mineral components, with the order C_3_A > C_4_AF > C_3_S > C_2_S [24], leading to differences in the hydration activity of various minerals [25]. H.W. Tian et al. studied the effect of superplasticizers on the flowability and early hydration of silicate cement and calcium sulfoaluminate cement systems and demonstrated that PCE22 with a high polycarboxylate ether (PCE) to cement ratio can significantly inhibit the early formation of AFt crystals in SAC pastes [1].

Han Jing [26], Han Fanghui [27], Li Zhiping [28], Hui Zhang [29], and others used the Krstulovic–Dabic model to study the hydration kinetics of cement or composite binder systems. By investigating the effects of internal and external factors such as cement dissolution rate, ion concentration, hydration product nucleation and crystal growth rate, and cement component composition on the rate and direction of hydration reactions, they examined the entire process of hydration reactions from a dynamic perspective [30]. However, there are currently no relevant reports on the effects of different additives on the hydration kinetics of cement.

The Krstulovic–Dabic model suggests that the hydration process of cement-based materials consists of three stages: nucleation and crystal growth (NG), interfacial reaction (I), and diffusion (D) [31]. The kinetic equations for these three processes are as follows (1)–(3):(1)$$\mathrm{NG}\;\;{\left[-\ln\left(1-\alpha \right)\right]}^{\frac{1}{n}}=K_{\mathrm{NG}}\left(t-t_{0}\right)$$
(2)$$I\;\;{\left[1-{\left(1-\alpha \right)}^{1/3}\right]}^{1}=K_{I}\left(t-t_{0}\right)$$
(3)$$D\;\;{\left[1-{\left(1-\alpha \right)}^{1/3}\right]}^{2}=K_{D}\left(t-t_{0}\right)$$
where α is the degree of hydration; n is the geometric growth index of crystals (J/g^−1^**)**; t is the hydration time, h; t_0_ is the end time of the induction period (h); and K_NG_, K_I_, and K_D_ are the reaction rate constants of the NG, I, and D processes, respectively.

The beginning of the acceleration period marks the start of the NG process. During the early stage of the acceleration period, with a sufficient water supply and high liquid-phase ion undersaturation, there are few hydration products, and the hydration rate is mainly controlled by the growth of C-S-H clusters (flocs) [16]. As the hydration progresses towards the end of the NG process, the limited growth space restricts the rapid migration ability of ions in the liquid phase, leading to a decrease in nucleation sites. At the transition point to the I or D process, the nucleation and growth rates reach their maximum points [32]. Therefore, the NG process can reflect the nucleation and growth of hydration products during the acceleration period. Y.R. Zhang et al. [33,34] believed that the maximum reaction rate is mainly determined by the total number of nucleations during the acceleration period. When there are many nucleations, the hydration products can form a dense covering layer at a small size, leading to a lower degree of cement hydration.

In this study, different PCEs with various molecular structures were synthesized to investigate the effects of PCE molecular structure on the dispersion performance, adsorption behavior, and hydration performance of different types of cement. The goal is to elucidate the basic principles of PCEs in different cement systems and establish a theoretical foundation for the development of high-performance PCEs with special functions.

## 2. Experimental Section

### 2.1. Materials

#### 2.1.1. Cement

In this study, P. I 42.5 RC from China United Cement Group Co., Ltd. (Beijing, China) and P.O 42.5 LC from Anhui Conch Group Co., Ltd. (Anhui, China) were used as the cementitious materials. The chemical composition and particle size distribution of the cement are shown in Table 1 and Table 2, respectively. The cement chemical composition analysis was conducted according to the Chinese standard GB/T 176-2017 [35], “Methods for chemical analysis of cement”. The cement particle size distribution was measured using a laser particle size analyzer (model: Winner 3000) from Jinan winner particle instrument stock Co., Ltd. (Jinan, China).

Based on the data results in Table 1, the content of cement mineral components was calculated according to the Chinese standard GB/T 21372-2008 [36] “Portland Cement Clinker”. The results are shown in Table 3.

The content of C_3_S, C_2_S, C_3_A, C_4_AF, and CaSO_4_ in the two types of cement was calculated based on the data in Table 1, and the results are presented in Table 3. Comparison of the mineral compositions of the two types of cement reveals that the content of C_3_S, C_3_A, C_4_AF, and CaSO_4_ in the RC is higher by 36.3%, 0.23%, 0.66%, and 1.1%, respectively, while the C_3_A content in the RC is lower by 1.1% compared to that in the LC.

#### 2.1.2. Synthesis

In this study, two types of polycarboxylate superplasticizers, PC-1 and PC-2, were synthesized in-house. PC-1 was prepared by esterifying methoxy polyethylene glycol (MPEG) with a molecular weight of 1200 with methacrylic acid (MAA) to obtain methoxy polyethylene glycol methacrylate (MPEGMA). Subsequently, it was copolymerized with acrylic acid (AA) under the action of initiators such as ammonium persulfate and sodium hypophosphite to obtain the target product. The copolymer was neutralized with sodium hydroxide, and the molar ratios of (MAA:MPEG) and (AA:MPEGMA) were 3.2:1 and 3.7:1, respectively. PC-2 was prepared by copolymerizing AA with isobutene-coupled polyethylene glycol ether (TPEG) under the action of initiators such as ammonium persulfate and sodium hypophosphite. The copolymer was neutralized with sodium hydroxide, and the molar ratio of (AA:TPEG) was 3.7:1. The difference between PC-1 and PC-2 lies in the molecular side chains. PC-1 has a molecular side chain of MPEGMAA with a molecular weight of 1200, while PC-2 has a molecular side chain of TPEG with a molecular weight of 2400. The chemical structures of PC-1 and PC-2 are shown in Figure 2a,b.

### 2.2. Polymer Characterization

#### 2.2.1. Gel Permeation Chromatography

The temperature was maintained at 25 °C, and a 0.1 mol/L NaNO_3_ aqueous solution with a pH of 7 was used as the eluent, with dextran of different molecular weights as the calibration standards. PCE was diluted to 5 mg/mL with a 0.1 mol/L sodium nitrate solution. GPC columns SB-804 HQ (size exclusion, 1,000,000 g/mol) and SB-802.5 HQ (exclusion volume, 10,000 g/mol) from Ohpak Shodex in Tokyo, Japan, were used for GPC measurements.

#### 2.2.2. Surface Charge Density Analysis

A particle charge detector, PCD 05 Smart, was used in combination with a standard titration solution to titrate and detect the surface charge density of particles.

### 2.3. Cement Slurry Experiment

#### 2.3.1. Slurry Flowability

Testing was conducted in accordance with the Chinese standard GB/T 8077-2012 [37] “Methods of testing for homogeneity of concrete admixtures”. The cement slurry mixer is shown in Figure 3, as depicted in the schematic diagram. The detailed composition of the evaluated slurry is shown in Table 4.

#### 2.3.2. Rheological Properties of Cement Slurry

Sample preparation: A cement slurry with a *w*/*c* ratio of 0.29 was prepared according to the method in GB/T 8077-2012 [37]. PCE was added to the cement slurry at a dosage of 0.25 mg/g cement. Rheological tests were conducted using a rotational rheometer (Rheolab QC, Anton Paar, Austria), with the prepared cement slurry immediately poured into a cylinder. The rotational rheometer is shown in Figure 4. The rotor diameter is 39 mm, the cylinder diameter is 42 mm, and the height of the rheological laminar flow is 1.5 mm.

Testing procedure: The instrument was monitored for 60 s to stabilize at 20 °C, then the sample was pre-sheared at a speed of 5 rad/min for 30 s to minimize the effect of thixotropy. The shearing rate was then increased from 0 to 131 rpm for 1 min, followed by holding at 131 rpm for 18 s. The shearing rate was then decreased from 131 rpm to 0 and increased again to 131 rpm for 1 min. The experimental temperature and humidity were maintained at 20 ± 2 °C and 60 ± 5%, respectively. Three tests were conducted for each type of cement slurry.

In this study, three models, including the Bingham model, the Modified-Bingham (M-B) model, and the Herschel–Bulkley (H-B) model, were used to fit the rheological parameters [38]. The three models are shown in Equations (4)–(6), respectively.
(4)$$\tau =\dot{\tau }+\eta \gamma \;(\mathrm{Bingham}\;\mathrm{model})$$
(5)$$\tau =\dot{\tau }+u\gamma +{a\gamma }^{2}\;(M\text{-}B\;\mathrm{model})$$
(6)$$\;\tau =\dot{\tau }+{k\gamma }^{n}\;(H\text{-}B\;\mathrm{model})$$
where τ is shear stress (Pa); $\overset{\cdot }{\tau }$ is yield stress (Pa); η and u are plastic viscosity (mPa·s); k is consistency factor; γ is shear rate (s^−1^); and n is the flow behavior index.

#### 2.3.3. Adsorption Amount

The adsorption amount was measured using an Eluent TOC-VCPH instrument. Different concentrations of water-reducing agent solutions were prepared, and 20 g of cement was added to 40 mL of the water-reducing agent solution. After stirring evenly, an appropriate amount of liquid was taken out and poured into a centrifuge tube. The upper clear liquid was collected after centrifugal filtration (at 5000 r/min for 10 min), and the TOC was tested. The adsorption amount of polycarboxylate superplasticizer on cement particles was calculated according to Formula (7).
(7)$$г=\frac{\left(c_{1}-c_{0}\right)\times V_{1}}{m}$$
where г is the adsorption amount of the water-reducing agent in the cement slurry (mg∙g^−1^); c_0_ is the total organic carbon content of the solution containing the water-reducing agent added to the cement slurry (g∙L^−1^); V_1_ is the volume of the water-reducing agent sample solution added in the experiment (mL); c_1_ is the total organic carbon content of the water-reducing agent sample solution (mg/L); and m is the mass of the cement sample (g).

#### 2.3.4. Hydration Heat

A TAM Air eight-channel isothermal microcalorimeter was used to monitor the heat-release rate of cement hydration in real-time. The experimental materials were placed in the same environment 24 h before the test, and the instrument was calibrated and balanced at the desired temperature. A water-to-cement ratio of 0.29 was used, and 100 g of cement and a diluted admixture solution were weighed and thoroughly mixed. The mixture was then weighed at 3.0 g and poured into a 20 mL ampoule, sealed, and placed in the insulated channel with the stopper tightly sealed for testing.

In order to better analyze the influence of PCE with different molecular structures on the hydration heat performance, t_0_, t_1_, t_2_, t_3_, and t_4_ represent the end time of the induction period or the start time of the NG process, the start time of the I process, the time at which the maximum slope of the acceleration period curve occurs, the end time of the acceleration period, and the start time of the D process, respectively. q_0_, q_2_, q_3_, Q_0_, Q_3_, and Q_0–3_ represent the corresponding heat-release rate and cumulative heat release at each point, and K_2_ is the transverse slope between 0 and 2, representing the nucleation rate during the acceleration period, as shown in Figure 5.

#### 2.3.5. Hydration Termination

To terminate the hydration process before observing hydration morphology, 1 g of the sample was placed in a 50 g bath of cold isopropanol (5 °C) at the desired testing time point to extract water from the sample. After stirring for 1 min, the mixture was soaked for 2 h. The cement-isopropanol suspension was then placed in a vacuum filtration funnel containing two layers of filter paper (0.45 μm) to filter the mixture. The filtered sample was then dried for 3 days in a vacuum dryer. For later termination, at each respective age, the test block was crushed to obtain the core part (small pieces of 4 mm × 4 mm and 2 mm) and washed with isopropanol 3 times before soaking for 2 h. The cement block-isopropanol soaking solution was then placed in a vacuum filtration funnel containing two layers of filter paper (0.45 μm) to filter the mixture. The filtered sample was then dried for 3 days in a vacuum dryer.

#### 2.3.6. Morphological Analysis

The sample obtained after the termination of hydration was affixed to a copper sample holder using a conductive adhesive, then vacuum-coated with gold, and observed for microstructure using scanning electron microscopy on the cross-section of the sample.

The morphology of the hydration products was characterized using a scanning electron microscope (SEM) with a resolution of 100,000 times under ambient conditions at a temperature of 25 ± 1 °C. The SEM used was a Korean COXEM-20.

## 3. Results and Discussion

### 3.1. Structural Characterization

The molecular structures of PC-1 and PC-2 synthesized were characterized by gel permeation chromatography and particle charge analysis, as shown in Table 5.

As shown in Table 5, the polymer components of PC-1 and PC-2 are 68.3% and 89.6%, respectively, indicating a higher conversion rate for PC-2. The weight-average molecular weight (Mw) of PC-1 and PC-2 was 14,356 g/mol and 44,843 g/mol, respectively. The hydrodynamic radius (Rh) of PC-1 and PC-2 was 2.85 and 5.10, respectively, indicating that the molecular weight and volume of PC-1 were smaller than those of PC-2 and that the charge density of PC-1 was higher than that of PC-2. This is mainly because the side chain of PC-1 is made by methoxy polyethylene glycol (MPEG) and methyl acrylate with a molecular weight of 1200. Excessive methyl acrylate is introduced into the copolymerization, and part of it participates in the copolymerization, resulting in a lower conversion rate in the copolymerization reaction. The carboxyl group content in PC-1 is higher than that in PC-2, resulting in a higher charge density in PC-1. Harding [39] proposed a conformation triangle, where a value of 0 indicates a close-packed sphere, a value of 0.5–0.8 indicates an unbranched structure, and a value of 1.8 indicates a rigid rod-like structure. Therefore, it can be inferred that the PC-1 molecular structure is between a “close-packed sphere and unbranched structure”, and the PC-2 molecular structure is closer to a “close-packed sphere”, indicating that the PC-1 molecule is more extended than PC-2.

### 3.2. Dispersion Performance

#### 3.2.1. Flowability

Using RC and LC, the net slurry flow rate was tested at 0.1%, 0.15%, 0.2%, 0.25%, and 0.3% for PC-1 and PC-2 folded solids admixtures, respectively, and the test results are shown in Figure 6.

As shown in Figure 6, with the increase in PC-1 and PC-2 dosages in both types of cement, the fluidity of the cement pastes gradually increased. The growth rate was more significant when the dosage of the superplasticizer was less than 0.2%, while the growth rate decreased when the dosage exceeded 0.2%. This is mainly because at higher dosages, the adsorption on the cement particle surface reaches saturation, and further increasing the dosage of the superplasticizer has little effect on the slump flow. In both RC and LC, the slump flow of PC-1 is superior to PC-2. This is mainly attributed to the higher charge density and better comb-like molecular structure of PC-1 compared to PC-2, as well as its smaller side-chain molecular weight and volume, which allow more molecules to adsorb on the cement particle surface. This leads to better dispersion between cement particles and hinders the formation of flocculated structures [40]. Moreover, the slump flow of both types of PCE in LC is superior to that in RC, indicating that PC-1 exhibits better dispersibility and that PCE also shows good dispersibility in LC. This is mainly because the LC contains a higher content of C_2_S than the RC. C_2_S particles in LC have higher reactivity, enabling them to react more quickly with water and form hydration products, thereby promoting cement dispersion. Additionally, LC has a smaller specific surface area than RC, resulting in a lower saturation point and a thicker water film on the cement particles. This hinders the formation of flocculated structures, leading to increased slump flow [41]. The adsorption of PCE molecules on the cement particle surface is related to the dispersing effect of PCE [42], so further analysis of the adsorption of PCE on the surface of the two types of cement particles was conducted.

#### 3.2.2. Rheological Properties

The internal structure of cement paste can be characterized by its rheological properties, and rheological parameters are important indicators for the workability and performance of fresh cement paste. The changes in the internal structure of the paste can be analyzed through rheological parameters such as shear rate, shear stress, and viscosity. In this paper, the effect of different molecular structures of water reducers on the rheological properties of cement paste in both the benchmark and LC was studied. The water–cement ratio was 0.29, and the dosage of the water reducer was 0.2%. The influence of shear rate on shear stress and apparent viscosity is shown in Figure 7a,b. The shear hysteresis loop curves and thixotropic hysteresis area of different water reducers in two types of cement slurries are presented in Figure 8a,b. To further investigate the effect of PCE molecular structure on the rheological properties of the paste, Bingham, Herschel–Bulkley [43], and Modified-Bingham models were used to fit the rheological data, as shown in Figure 9a–c and Table 6.

According to Figure 7, it can be seen that the shear stress of different water reducers in the standard cement and LC increases with the increase in shear rate, indicating non-Newtonian fluid characteristics. Under the same water reducer condition, the shear stress and apparent viscosity of the standard cement are both higher than those of the LC, indicating that the standard cement slurry is thicker than the LC slurry. Under the same water reducer conditions, the shear stress and apparent viscosity of the standard cement are higher than those of LC, indicating that the standard cement paste is thicker compared to LC. This is mainly attributed to the higher C_2_S content in LC, which results in fewer hydration products during the cement hydration process, leading to a decrease in the cohesive forces and viscosity of the cement paste. Additionally, LC has a smaller specific surface area, resulting in a thicker water film and fewer flocculated structures, leading to lower shear stress and apparent viscosity. Under the same cement conditions, the shear stress and apparent viscosity of PC-2 are higher than those of PC-1, indicating that the addition of PC-2 results in a thicker cement paste with a higher amount of flocculated structures and increased frictional resistance between cement particles. As shown in Figure 8, it can be observed that the hysteresis area of the thixotropic loop is smaller for the slurry using LC and PC-1. Consequently, the degree of flocculation is reduced, and its structure is more susceptible to disruption. This indicates that the use of LC and PC-1 can greatly improve the rheological properties of the cement slurry, which has a good correlation with the flowability results and follows the same trend.

As shown in Table 6, it can be observed that three models were used to fit the rheological curves, and the correlation coefficients were all above 0.99, indicating a high degree of correlation. However, when fitting the Herschel–Bulkley and Modify-Bingham models, negative values were obtained for the yield stress, indicating that these two models are not suitable for fitting. From the data fitted by the Bingham model, it is evident that under the same water reducer conditions, LC can significantly reduce the yield stress of the cement slurry by 57.5% to 80.7% compared to RC. Under the same cement conditions, PC-1 can significantly reduce the yield stress of the cement slurry by 27.8% to 67.2% compared to PC-2. Therefore, both LC and PC-1 can greatly reduce the yield stress of the slurry, thereby improving the dispersibility of the cement paste.

### 3.3. Adsorption Behavior

The adsorption behavior of PC-1 and PC-2 on RC and LC was studied using TOC tests. The relationship between the adsorption amount of PCE on the surface of cement particles and its dosage is shown in Figure 10.

As shown in Figure 10, it can be observed that, for the same cement comparison, the adsorption amount of PC-1 is significantly greater than that of PC-2 in both CC and LC. This is mainly attributed to the higher charge density and better comb-like molecular structure of PC-1 compared to PC-2, as well as the smaller molecular volume. These factors allow the negative charges in the main chain of PC-1 to adsorb on the surface of the silicate particles without being hindered by spatial resistance [44,45]. Consequently, PC-1 exhibits greater adsorption on the cement particle surfaces [9,46], leading to a higher adsorption amount than PC-2. This observation correlates well with the results of the flowability tests. Under the same PCE conditions, the adsorption amount in RC is higher than that in LC. This is mainly because the C_3_S content in RC is much higher than that in LC, and the C_2_S content in LC is higher than that in RC. PCE adsorbs more C_3_S than C_2_S [47], so the adsorption amount in RC increases, indicating that the C_3_S content in cement has a significant impact on PCE adsorption. In addition, RC has a larger specific surface area, providing adsorption sites, and its surface is negatively charged. Ions undergo physical adsorption by electrostatic attraction [48]. The larger the surface free energy of the RC system, the less thermodynamically stable it is. The physical or chemical reactions generated by the adsorption of water reducers on cement particles or hydration products can reduce the surface free energy of the system, resulting in a larger adsorption amount in RC.

To further determine the characteristic adsorption platform of polycarboxylate ether (PCE), Langmuir, Freundlich, and Dubinin–Radushkevich adsorption models were used to fit the differences in the adsorption performance of different PCEs. The results are shown in Table 7.

As shown in Table 7, the correlation coefficients obtained from Langmuir and D-R adsorption models were both higher than 0.98, except for the RC sample PC-2. The correlation coefficient obtained from the Freundlich adsorption model was lower, indicating that PCE adsorption on cement belongs to monolayer adsorption [49,50]. Compared to the Freundlich model, the D-R model had a better fitting effect. Dubinin et al. proposed the micropore filling theory, which introduced adsorption potential into the theoretical research of microporous adsorption. The D-R adsorption potential theory did not assume that the adsorbent surface was uniform and had a constant adsorption potential. Therefore, the D-R isotherm model has a wider applicability range than the Langmuir model, and the adsorption process can be distinguished as physical or chemical adsorption by the D-R model [51].

However, comparing the saturation adsorption amounts q^e^ and q_m_ obtained from Langmuir and D-R adsorption models, the q_m_ obtained from the D-R model was closer to the experimental values, indicating that the D-R adsorption model was more suitable for analyzing the isothermal adsorption of water reducers. Moreover, for the same water reducer, the saturation adsorption amount q_m_ of the RC was greater than that of the LC. For the same cement, the q_m_ of PC-1 was greater than that of PC-2, which is consistent with the law that the adsorption amount increases with the increase in PCE usage.

E is the adsorption-free energy, which can determine the nature of adsorption. E ranging from 8 to 16 KJ∙mol^−1^ belongs to ion exchange; E < 8 KJ∙mol^−1^ belongs to physical adsorption; and E > 16 KJ∙mol^−1^ belongs to chemical adsorption. The adsorption-free energy E of PC-1 and PC-2 in both types of cement is less than 8 KJ∙mol^−1^ [52,53,54,55]. Partially hydrolyzed polyacrylate (PCE) superplasticizers are gradually wrapped by hydration products, resulting in a decrease in concentration, and the concentration difference as a driving force decreases. Therefore, the main adsorption mechanism of PC-1 and PC-2 in both types of cement is physical adsorption. The PCE molecules are attracted to the charged points on the surface by electrostatic attraction or chemical bonding force. As cement hydration progresses, PCE will react with calcium ions in the solution and adsorb onto the surface of hydration products such as Aft or AFm [42,56,57].

### 3.4. Hydration Properties

#### 3.4.1. Hydration HEAT

The hydration heat-release rate curve and the cumulative heat-release curve are shown in Figure 11a,b, and characteristic parameters are extracted from the hydration heat-release rate curve as shown in Table 8.

During the induction period, the mineral phases gypsum and C_3_A dissolve rapidly, and AFt is formed. Meanwhile, the concentrations of Ca^2+^, OH^−^, SO_4_^2−^, and other ions in the cement solution continue to increase. When the ion concentrations reach a certain value, nucleation and crystallization occur at the solid–liquid interface, resulting in the formation of calcium hydroxide (CH) and ettringite (AFt) crystals. Comparing the C_3_A content, it can be inferred that the amount of AFt produced in the LC is higher than that in the RC.

According to Table 8, during the induction period, the time t_0_ for the LC to end its induction period is higher than that for the RC. At the end of the induction period, the hydration rate q_0_ for the LC is lower than that for the RC, indicating that the degree of delay in hydration of the LC without adding PCE is greater than that of the RC. Adding PCE with different structures has a significant impact on the delay of cement hydration. This is because PCE has different adsorption capabilities on cement particles. Compared with PC-2, PC-1 has a higher molecular charge density, a smaller molecular volume, and a better comb-like molecular structure, which can improve its adsorption capacity. It has a strong chelation effect on the Ca^2+^ on the surface of C_3_S particles, which can reduce the Ca^2+^ ion concentration in the solution, inhibit the crystallization of Ca(OH)_2_, reduce the formation of C-S-H gel, and thus delay the hydration of cement more than PC-2 [58]. This is manifested as an increase in t_0_ at the end of the induction period and a decrease in the hydration rate q_0_ for PC-1. However, after adding PCE, the degree of delay in hydration of the RC is greater than that of the LC, which is the opposite of the situation without PCE. This is because the C_3_S content of the RC is much higher than that of the LC, and PCE has a larger adsorption capacity in the RC, which can inhibit the dissolution of C_3_S and block some C-S-H nucleation [59,60,61]. This manifests as a larger t_0_ for the end of the induction period of the RC.

During the acceleration period, hydration products rapidly nucleate and grow, and K_2_ represents the maximum slope point of the hydration heat-release curve, indicating the nucleation rate. The larger the K_2_, the faster the nucleation rate. q_3_ represents the highest point of the hydration heat-release curve, indicating the nucleation amount. The larger the q_3_, the greater the nucleation amount. Q_0–3_ represents the accumulated heat during the acceleration period, indicating the degree of hydration. The larger Q_0–3_, the greater the degree of hydration [62].

Compared to the RC, both K_2_ and q_3_ of the LC are larger, indicating that the nucleation rate and nucleation amount of the LC are greater. This is mainly because the C_3_S content of the RC is higher during the acceleration period, which can neutralize a large number of Ca^2+^ ions in the pore solution, thereby delaying the crystallization and nucleation of Ca(OH)_2_ [33].

The lower C_3_S content of the LC and the lower degree of PCE encapsulation on the LC particles lead to more nucleation sites, which can form more fine hydration products. The hydration products come into contact with each other to form a dense hydration product coating, which slows down the release of calcium ions and delays the hydration process during the acceleration period, resulting in a larger t_3_ value for the LC compared to the RC. The LC particles are wrapped by fine hydration products, reducing the surface porosity of the cement particles, and C-S-H blocks the entry of pore solution, resulting in a lower degree of hydration for the LC at the same time. Comparing the q_3_ values of different PCEs added to the same cement, the differences are small. It is inferred that different molecular structures, PCE-1 and PCE-2, have little impact on the nucleation of hydration products in the later acceleration period [63]. However, the Q_0–3_ values of the samples with PCE added are greater than those without PCE, indicating that well-dispersed cement particles with PCE added can come into contact with water more, making the acceleration period of hydration more complete.

#### 3.4.2. Hydration Kinetics

In this paper, the Krstulovic–Dabic model was used to establish a hydration reaction kinetics model of cementitious systems based on isothermal microcalorimetry. The parameter changes of the kinetic model were investigated under different conditions of water reducers and cement, and a relationship function between the reaction hydration degree and hydration rate was established. By fitting the hydration process of a single cement system, the impact of PCE on hydration products and different cement hydration mechanisms and control factors in different hydration stages were revealed. The α-dα/dt curves of the hydration reaction for blank samples of ordinary RC and seashell RC (SPC), PC-1, and PC-2 in the two types of cement are shown in Figure 12a–f, and the summary of the hydration kinetics parameters of each sample is shown in Table 8. From the hydration heat-release rate curve, t_0_, t_1_, and t_4_ are the start times of the NG process, I process, and D process, respectively. The specific parameters are shown in Table 9. Combined with the hydration kinetics, it can be known that the NG, I, and D processes are all in the acceleration stage.

As shown in Table 9, it can be observed that the values of Qmax vary among different samples. The reaction rate constant K_NG_ for the NG process in all samples is much larger than the constants K_I_ and K_D_ for the I and D processes. This is because the early hydration of cement belongs to a self-catalytic reaction, and the hydration rate of the NG process is much faster than that of the I and D processes [26].

K_NG_ is the rate constant for the nucleation and crystal growth of the NG process. Except for RC-PC-1, the K_NG_ of LC is greater than that of ordinary RC in samples without PCE or with the same amount of PCE added. This is because the crystallization and growth of C-S-H mainly occur in the NG stage, and the nucleation and growth rates are related to the ion concentration in the solution. However, the high C_3_S content of ordinary RC neutralizes a large amount of Ca^2+^ ions in the pore solution, leading to a high degree of undersaturation of the liquid phase ions and delaying the crystallization and nucleation of Ca(OH)_2_ [33], which is consistent with the results of K_2_ analysis of the heat of hydration. n represents the degree of concentration’s impact on the reaction rate, and a larger n indicates a slower reaction [64,65]. When hydration progresses to the end of the NG process, the lack of growth space limits the rapid migration ability of ions in the liquid phase, causing the nucleation sites to decrease, as shown in Figure 8. Compared with samples without PCE, the n value of LC is comparable to that of ordinary RC, and the n value of LC is greater than that of ordinary RC after the same amount of PCE is added. Under the same cement conditions, the n value of PC-2 is greater than that of PC-1, indicating that the influence of LC and PC-2 on the rate of nucleation and crystal growth in the NG process is significant. Therefore, the growth rates of the I and D processes of LC and PC-2 decrease significantly, and their n values are both greater than 2. The crystals are mainly thin, plate-like C-S-H [29].

Under the same PCE conditions, LC and PC-2 added under the same cement conditions have smaller K_I_ values, indicating that the hydration degree of LC is lower than that of ordinary RC, and adding PC-2 can further reduce the hydration degree, as evidenced by the size of α_1_. This indicates that a large number of ions in the I process pass through the particle and hydration product boundaries of cement. LC, compared with ordinary RC and PC-2 compared with PC-1, generates more C-S-H gel precipitation on the particle surface during hydration, making ion migration difficult [32,33,34]. The dispersibility of PCE in LC is better than that in ordinary RC, and the sufficient water content results in a higher hydration rate [66], consuming more Ca^2+^ ions in the pore solution, which is the main reason for the decrease in the reaction rate at the phase boundary and diffusion stage.

As hydration progresses, the thickness of the hydration product layer increases, and the reaction gradually becomes difficult. The hydration control factor begins to shift towards the diffusion reaction rate D [16,24]. As shown in Figure 8, the D process fits well with the later stage of the hydration acceleration period, and K_D_ decreases almost by an order of magnitude compared to K_I_, indicating that inter-particle diffusion does not require liquid participation. Entering the D process sharply reduces the porosity and permeability of the slurry, and the C-S-H layer with very low permeability covers the Ca(OH)_2_ crystals and unhydrated particles. This greatly increases the diffusion resistance of water, Ca^2+^, and OH- approaching the unreacted cement particles, resulting in a low reaction rate during the D process [27]. In the D process, the reaction rates of the blank sample and the sample with added water reducer are basically the same, indicating that the addition of PCE has a small impact on the D process.

α_1_ is the boundary point where the NG process transitions to the I process. After adding PCE, the baseline cement α_1_ is greater than that of the LC, indicating that after adding PCE, the hydration reaction only transitions from the NG process to the I process when the baseline cement has a higher degree of hydration [32].

#### 3.4.3. Morphological Analysis

The morphologies of blank samples and 12 h hydration products of both RC and LC with the addition of PC-1 and PC-2 are shown in Table 10. The morphological characteristics of hydration products and their corresponding degrees of hydration are shown in Table 11.

At 12 h of hydration, as shown in Table 10 and Table 11, the blank sample is at the beginning of the D process, while the other PCE samples are at the beginning of the I process. From the degree of hydration, it can be observed that the blank sample has a significantly higher degree of hydration compared to the samples with added PCE, resulting in differences in morphology. According to Table 9, the blank sample exhibits a noticeable fibrous C-S-H structure, while the PCE-added samples show unhydrated C_3_S particles, indicating that the addition of PCE inhibits the early hydration of C_3_S, resulting in fewer hydration products [56]. The differences between the samples are minimal, and they are all in the rapid nucleation stage.

At 1 day of hydration, the degree of hydration for PCE samples is slightly lower than that of the blank sample, and they both enter the D stage, indicating that PCE delays the mid-late-stage hydration of cement to some extent but with minimal impact. For the same PCE, the hydration products of the RC exhibit a needle-like morphology with lengths of approximately 1~2 μm, while the hydration products of the LC show either a length of approximately 1 μm or no apparent needle-like structures, consistent with the lower degree of hydration in the LC, suggesting the inhibition of hydration reactions. For the same cement, the addition of PC-1 results in a higher presence of needle-like hydration products, while PC-2 exhibits predominantly fibrous products, consistent with the lower degree of hydration in PC-1, indicating a higher degree of hydration compared to PC-2. This observation is consistent with the kinetics of hydration analysis, as the K_NG_ and K_I_ values for PC-1 are both higher than those of PC-2.

At 3 days of hydration, the hydration degree of the PCE-added cement was still lower than that of the blank sample, with a smaller decrease than at 12 h of hydration. From the hydration kinetics curves, it can be seen that at 3 days, both the blank sample and the PCE sample entered the D plateau stage, and the hydration heat curves were also in the steady state. Therefore, it can be inferred that PCE to some extent inhibited the later hydration of cement, but the degree of influence was not significant. Under the same cement conditions, after adding PC-1 and PC-2, the morphologies of the hydration products were similar, with both being gel substrates accompanied by some voids and needle-like hydration products that extended between the gaps of the particles, bridging them; there were also rod-like and plate-like hydration products.

#### 3.4.4. Mechanism of Action

The mechanism of action of the RC and LC slurry under PCE was deduced through hydration kinetics and morphology analysis. As shown in Figure 13, the hydration products layer on the particles in the NG process is thin, while the hydration products (C-S-H and CH) nucleate on the surface and near the surface of the hydrated particles, and a small amount of AFt exists in the solution. The content of C_3_S in the RC is much higher than that in the LC, and PCE has a greater adsorption capacity in the RC, which can inhibit the dissolution of C_3_S and block some C-S-H nucleation. Therefore, the C-S-H content in the LC is higher than that in the RC, and the dispersibility of the LC is better than that of the RC, making the cement particles better dispersed and the water content sufficient.

In the I process, the superplasticizer on the surface of the cement particles is wrapped by the hydration layer and loses its effect, while a small amount of superplasticizer in the pore solution is adsorbed on the surface of the hydration products, providing more space for the growth of the hydration products layer. The hydration products layer gradually becomes thicker, while the LC generates a thicker hydration products layer in the NG process, which hinders the generation of new C-S-H gel in the I process and consumes more Ca^2+^ in the pore solution. The spacing between the hydration products is greater than that in the RC.

In the D process, some needle-like hydration products are wrapped by amorphous C-S-H, and the hydration product layer also spreads inside the cement particles. Ca^2+^ and sulfate ions in the pore solution are nearly consumed, and the hydration products are in close contact, with the slowest diffusion rate as the porosity decreases.

## 4. Conclusions

By comparing the dispersion, adsorption, and hydration performance of PCE with different molecular structures in ordinary RC and LC, the following research results were obtained:

(1) Based on the slurry and rheological properties, it can be observed that PC-1 exhibits a higher charge density compared to PC-2, with a more favorable molecular structure characterized by better combing of side chains, a lower side-chain molecular weight, and a smaller molecular volume. These characteristics allow PC-1 to adsorb more on the surface of cement particles and improve the dispersion, resulting in a reduction in the slurry’s yield stress by more than 27.8%. On the other hand, the high C_2_S content and smaller specific surface area of the LC contribute to the reduction in flocculation structures and a decrease in the slurry’s yield stress by more than 57.5%, indicating its good flowability. Therefore, the use of LC and PC-1 significantly enhances the rheological performance of cement slurries and correlates well with the flowability results.

(2) From the adsorption performance, it was found that the negative charge in the main chain of PC-1 is not affected by the steric hindrance of the side chain, which makes the adsorption amount of PC-1 much larger than that of PC-2, consistent with the dispersion results. PCE has a larger adsorption amount on C_3_S, resulting in a larger adsorption amount in ordinary RC with a higher C_3_S content than in LC. The D-R model can better reflect the actual test adsorption amount.

(3) From the hydration heat, it was found that the adsorption amount of PC-1 on cement particles is greater than that of PC-2, which delays the induction period of hydration for PC-1. Different PCEs have a small influence on nucleation during the acceleration period. Under the same PCE conditions, the adsorption amount of PCE is larger in ordinary RC, which can inhibit C_3_S dissolution and block part of the C-S-H nucleation, resulting in a greater delay of the induction period in ordinary RC than in LC. The higher C_3_S content in ordinary RC can neutralize a large amount of Ca^2+^ ions in the pore solution, thus delaying the crystallization and nucleation of Ca(OH)_2_ during the acceleration period. The LC is wrapped in small hydration products, resulting in a lower degree of hydration at the same time.

(4) From the hydration kinetics, except for RC-PC-1, the K_NG_ of LC is larger than that of ordinary RC, indicating that LC promotes nucleation and growth during the early acceleration period, consistent with the analysis results of K_2_ and q_3_ in hydration heat curves. Due to the precipitation of C-S-H gel generated by LC and PC-2 on the surface of particles, ion migration becomes difficult, resulting in a decrease in K_I_ and hydration degree α_1_, which inhibits hydration. In the D process, the K_D_ of the blank sample and the sample with added water reducer are basically the same, indicating that the addition of PCE has little effect on the D process.

(5) From the morphological analysis, it can be observed that at 1 day of hydration, with the same PCE, the hydration products of the RC exhibit a needle-like morphology with a length of approximately 1~2 μm, while the hydration products of the LC show a length of approximately 1 μm or no apparent needle-like structures, consistent with the lower degree of hydration of the LC compared to the RC. For the same cement, the addition of PC-1 results in a higher presence of needle-like structures in the hydration products, while PC-2 predominantly exhibits fiber-like structures. This observation aligns with the lower hydration degree of PC-1 compared to PC-2, as well as the higher values of K_NG_ and K_I_ in the hydration kinetics of PC-1 compared to PC-2. It can be inferred that both the LC and PC-2 inhibit early-stage hydration. At 3 days of hydration, the addition of different structured PCE does not significantly affect the morphology of the hydration products.

## Figures and Tables

**Figure 1 materials-16-04168-f001:**
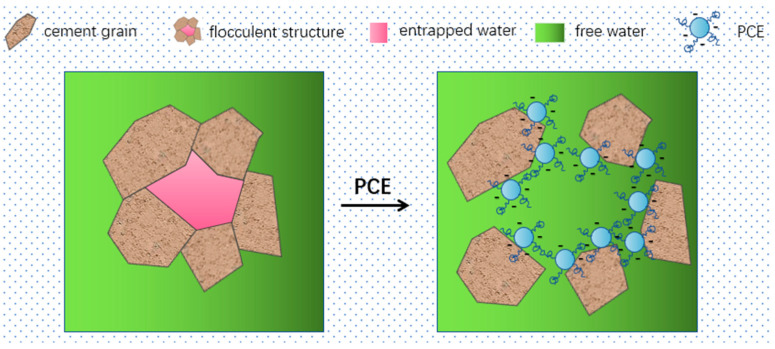
Schematic diagram of the dispersion mechanism of PCE in cement.

**Figure 2 materials-16-04168-f002:**
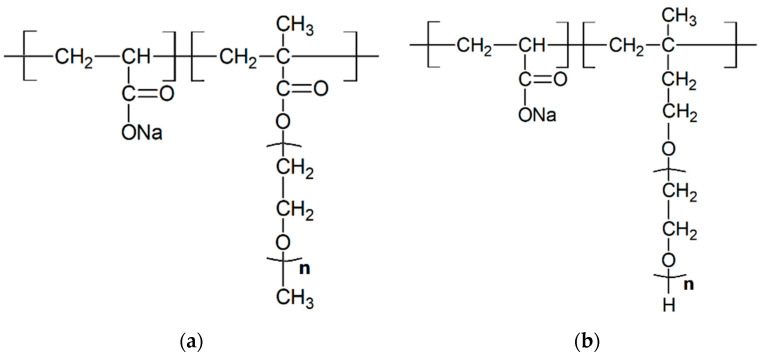
Chemical structure of PC-1 (**a**) and PC-2 (**b**).

**Figure 3 materials-16-04168-f003:**
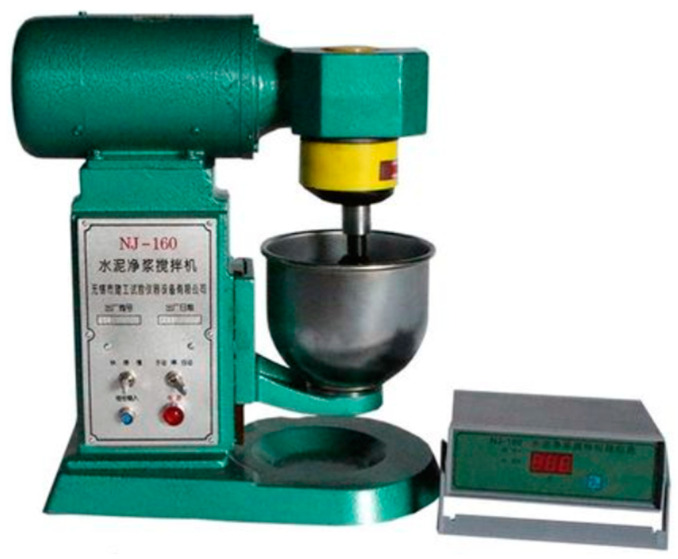
Cement slurry mixer.

**Figure 4 materials-16-04168-f004:**
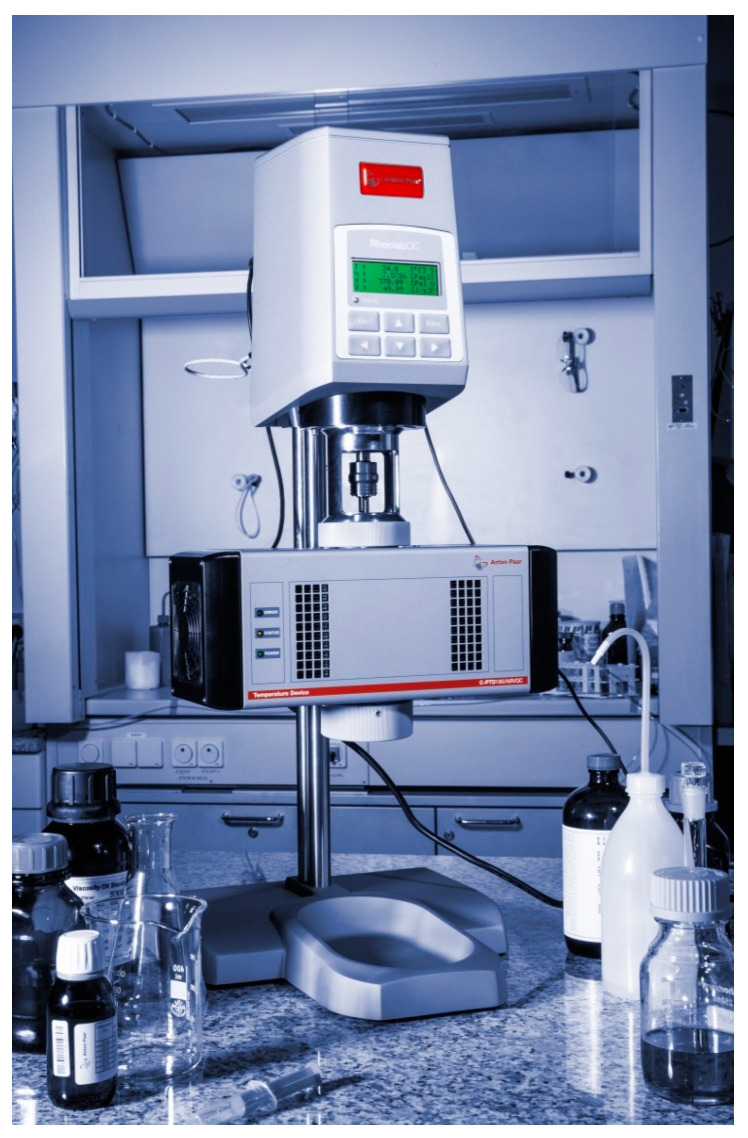
Rotational rheometer.

**Figure 5 materials-16-04168-f005:**
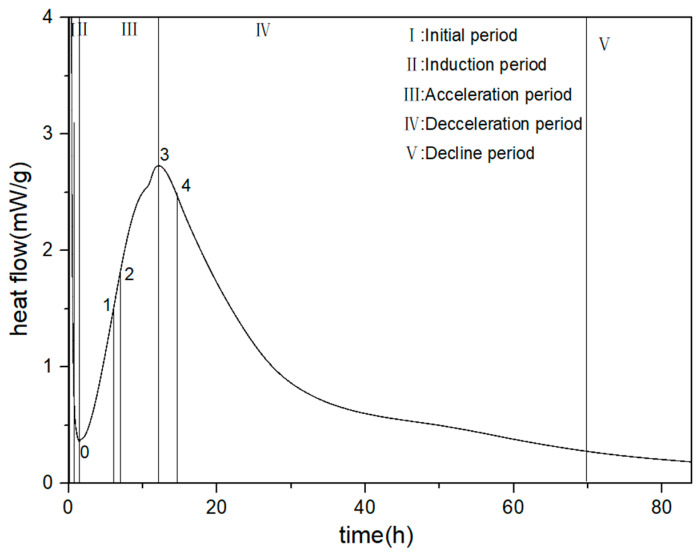
Five stages of hydration heat release.

**Figure 6 materials-16-04168-f006:**
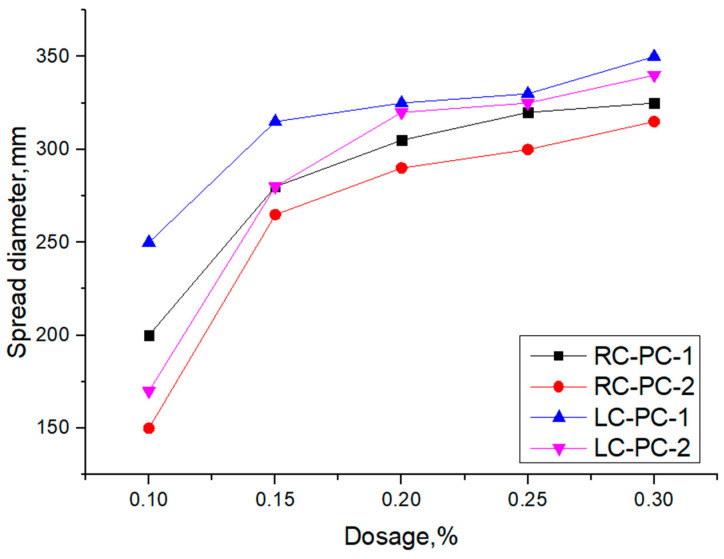
Influence of synthesized polymers on the fluidity of cement pastes.

**Figure 7 materials-16-04168-f007:**
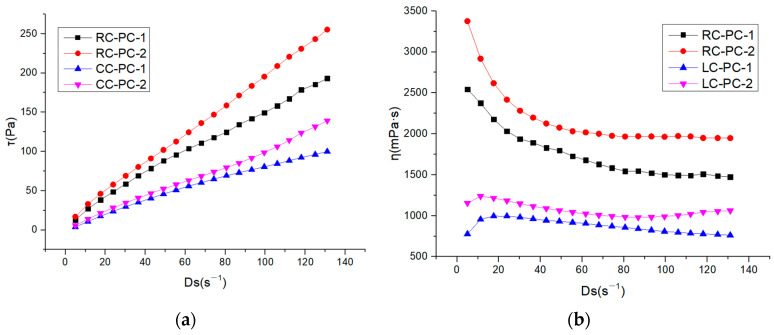
Rheological properties of different water reducers in two types of cement slurries: (**a**) Shear rate–shear stress; (**b**) Shear rate–apparent viscosity.

**Figure 8 materials-16-04168-f008:**
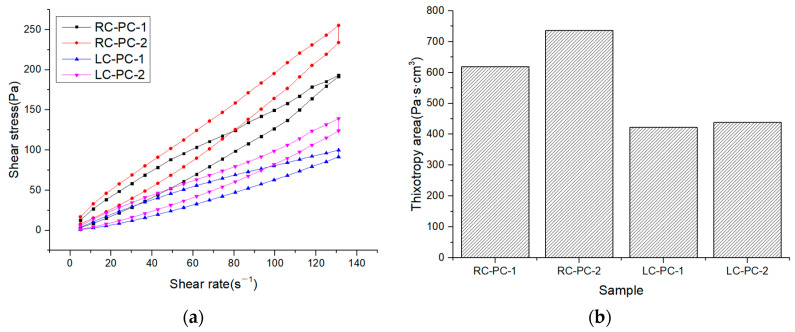
Shear hysteresis loop curves and thixotropic hysteresis area of different water reducers in two types of cement slurries: (**a**) Thixotropic loop; (**b**) Thixotropic hysteresis area.

**Figure 9 materials-16-04168-f009:**
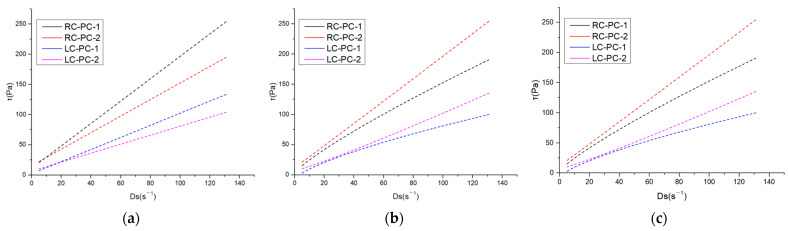
Three model-fitting curves: (**a**) Bingham; (**b**) Herschel–Bulkley; (**c**) Modify-Bingham.

**Figure 10 materials-16-04168-f010:**
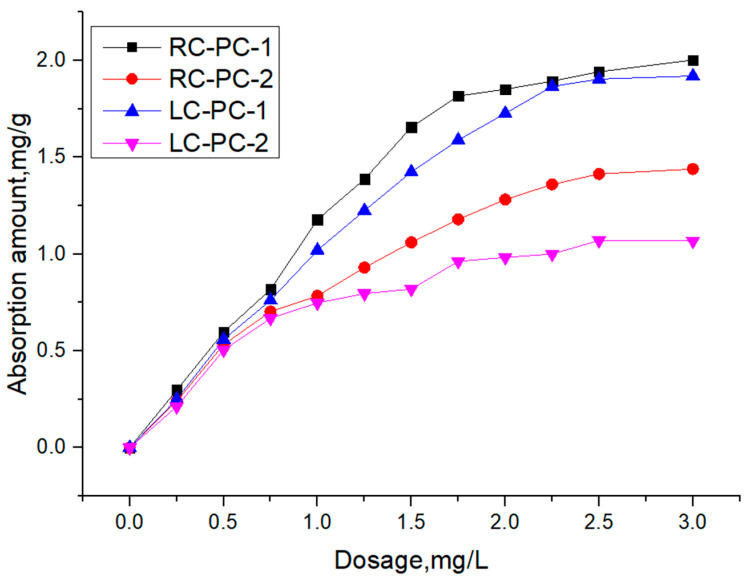
Adsorption amount of two types of PCE in two types of cement slurries.

**Figure 11 materials-16-04168-f011:**
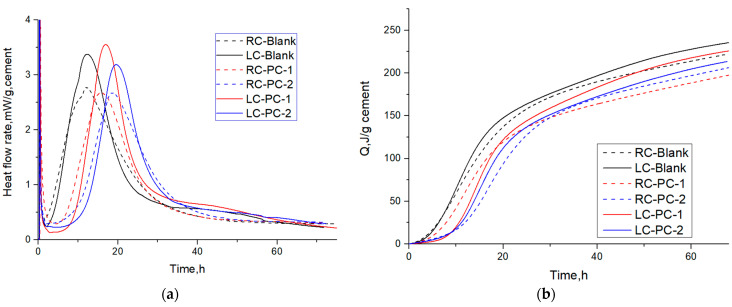
Heat flow and cumulative heat flow of different PCEs in two types of cement: (**a**) Heat flow; (**b**) Cumulative heat flow.

**Figure 12 materials-16-04168-f012:**
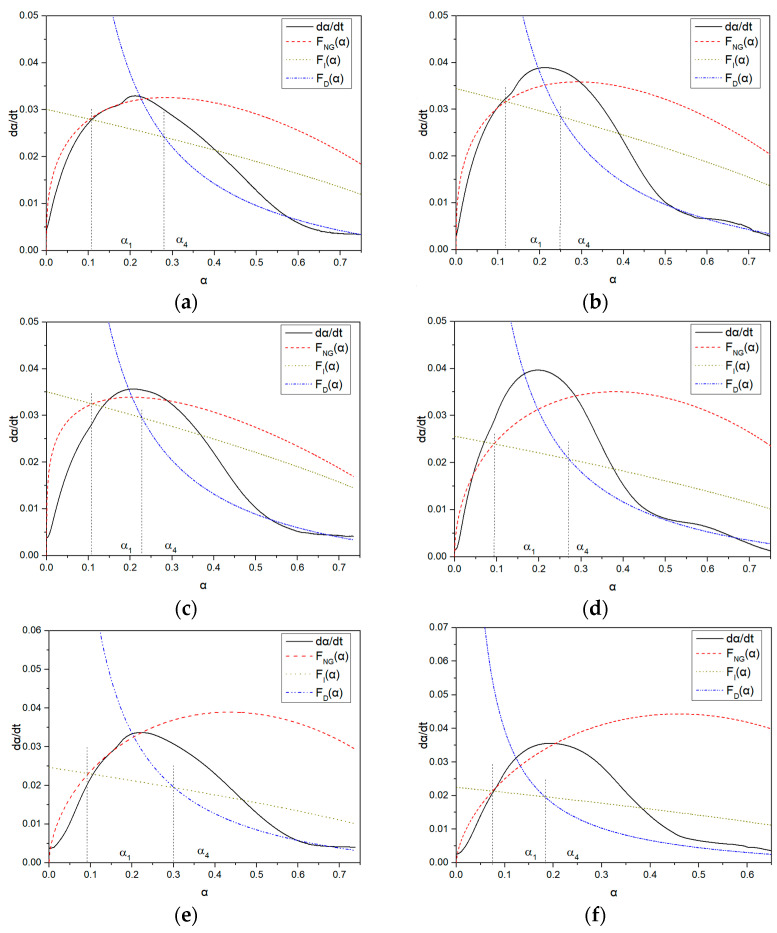
α-dα/dt curves of hydration reactions for different molecular structures of PCE in two types of cement: (**a**) RC-Blank; (**b**) LC-Blank; (**c**) RC-PC-1; (**d**) LC-PC-1; (**e**) RC-PC-2; (**f**) LC-PC-2.

**Figure 13 materials-16-04168-f013:**
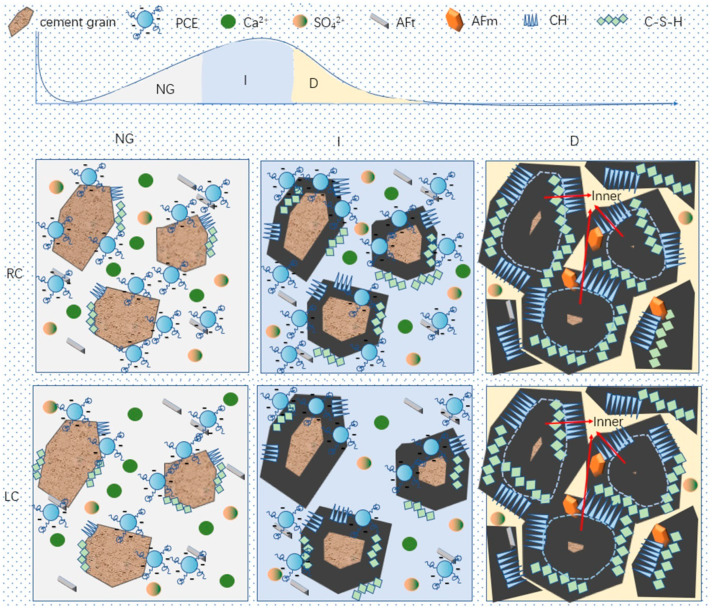
Schematic diagram of the hydration mechanism.

**Table 1 materials-16-04168-t001:** Chemical composition of the cement.

Cement	CaO	SiO_2_	Al_2_O_3_	Fe_2_O_3_	MgO	SO_3_	Na_2_O	K_2_O	MnO	TiO_2_	LOI *
RC	63.79	19.8	5.12	3.65	2.3	2.49	0.30	0.31	0.12	0.16	1.85
LC	58.98	21.72	5.60	3.53	2.55	2.10	0.31	0.38	0.15	0.14	4.00

* Loss of Ignition.

**Table 2 materials-16-04168-t002:** Particle size distribution of the cement.

Cement	X50 (μm)	<3 μm (%)	3~32 μm (%)	32~65 μm (%)	>65 μm (%)	>80 μm (%)	Specific Surface Area (m^2^/kg)
RC	14.581	18.903	67.724	13.229	0.144	0.000	355
LC	15.340	18.744	66.967	14.113	0.176	0.000	343

**Table 3 materials-16-04168-t003:** Mineral compositions of the cement.

Cement	C_3_S	C_2_S	C_3_A	C_4_AF	CaSO_4_
RC	62.78	9.40	7.60	7.25	4.23
LC	26.49	42.28	8.87	7.02	3.57

**Table 4 materials-16-04168-t004:** Composition of the evaluated slurry.

Composition	Cement	Water
Dosage/g	300	87

**Table 5 materials-16-04168-t005:** Characterization of PCE molecular structure.

Sample	Mw/(g∙mol^−1^)	Conversion/%	Rh	Mark–Houwink a	Charge Density (μeq/g)
PC-1	14,356	68.3	2.85	0.43	5529
PC-2	44,843	89.6	5.10	0.17	4393

**Table 6 materials-16-04168-t006:** Fitting results of Bingham, Herschel–Bulkley, and Modified-Bingham models for rheological curves of different PCE in two types of cement slurries.

Sample	Bingham			Herschel–Bulkley				Modify-Bingham			
	τ/ mPa	η/ (mPa·s)	R^2^	K/ (Pa·sn)	τ/ mPa	n	R^2^	μ	τ/ mPa	a	R^2^
RC-PC-1	11.173	1.8547	0.9995	3.185	2.7931	0.836	0.998	1.602	10.060	−0.0017	0.9974
RC-PC-2	15.477	1.3696	0.9957	1.766	12.008	1.010	0.999	1.807	12.285	0.0004	0.9995
LC-PC-1	2.153	1.002	0.9948	3.290	−7.503	0.715	0.999	1.028	−0.075	−0.0021	0.9994
LC-PC-2	6.571	0.7431	0.9900	0.586	6.684	1.106	0.995	0.834	6.057	0.0012	0.9964

**Table 7 materials-16-04168-t007:** Isothermal adsorption equation fitting parameters.

Sample	Langmuir			Freundlich			D-R			
	q^e^ (mg∙g^−1^)	K_L_ (L∙mg^−1^)	R^2^	K_f_ (mg^(1−1/n)^·L^1/n^/g)	n_f_	R^2^	k_D_/10^−6^ (kJ∙mol^−1^)	q_m_	E (kJ∙mol^−1^)	R^2^
RC-PC-1	3.6778	0.4597	0.9858	1.1212	1.6033	0.9652	0.2199	2.1988	1.59	0.9868
RC-PC-2	2.5018	0.4985	0.9947	0.8098	1.6727	0.9848	0.1710	1.5003	1.71	0.9665
LC-PC-1	3.6338	0.3992	0.9902	1.0076	1.5353	0.9737	0.1550	2.0407	1.55	0.9838
LC-PC-2	1.4675	0.9841	0.9884	0.6944	2.1548	0.9665	0.2100	1.0903	2.10	0.9837

**Table 8 materials-16-04168-t008:** Parameters of hydration heat-release curves of different PCEs in two types of cement.

Sample	t_0_ (h)	q_0_ (mW/g)	Q_0_ (J/g)	t_2_ (h)	K_2_ (mW/(g∙h))	q_2_ (mW/g)	t_3_ (h)	q_3_ (mW/g)	Q_3_ (J/g)	Q_0–3_ (J/g)
RC-Blank	1.68	0.36	2.92	5.6	0.38	1.38	12.15	2.77	80.41	77.49
LC-Blank	2.15	0.24	3.1	6.7	0.49	1.60	12.36	3.38	92.85	89.75
RC-PC-1	4.33	0.28	7.15	10.37	0.33	1.39	15.77	2.68	94.59	87.44
RC-PC-2	3.21	0.30	3.50	13.95	0.34	1.56	18.59	2.67	99.56	95.78
LC-PC-1	2.90	0.13	1.46	12.12	0.51	1.65	16.93	3.55	97.81	96.35
LC-PC-2	2.16	0.24	1.81	15.4	0.43	1.73	19.60	3.19	109.27	107.46

**Table 9 materials-16-04168-t009:** Hydration kinetic parameters of different molecular structures of PCE in two types of cement.

Sample	Q_max_/J.g^−1^	t_50_/h	NG		I	D	Hydration Degree		
			K_NG_ (h^−1^)	n	K_I_ (μm/h)	K_D_ (μm^2^/h)	α_0_	α_1_	α_4_
RC-Blank	303.03	24.24	0.0437	1.51	0.0100	0.0021	0.010	0.106	0.281
LC-Blank	312.50	21.97	0.0480	1.52	0.0115	0.0021	0.010	0.117	0.250
RC-PC-1	270.27	27.30	0.0461	1.30	0.0117	0.0019	0.027	0.108	0.227
RC-PC-2	285.71	27.29	0.0409	2.32	0.0082	0.0019	0.012	0.092	0.300
LC-PC-1	322.58	27.90	0.0420	1.92	0.0085	0.0017	0.004	0.095	0.269
LC-PC-2	322.50	33.58	0.0427	2.59	0.0075	0.0010	0.005	0.076	0.185

**Table 10 materials-16-04168-t010:** Morphological analysis of hydration products of RC and LC with different water reducers at 12 h and 3 d.

Cement	Hydration Time	Blank	PC-1	PC-2
RC	12 h	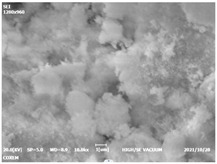	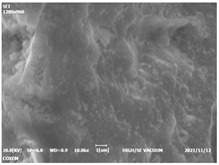	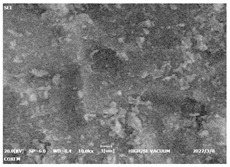
LC		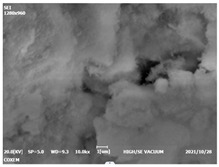	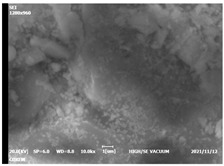	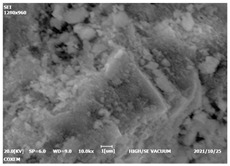
RC	1 d	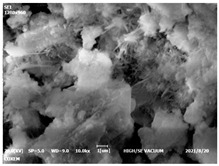	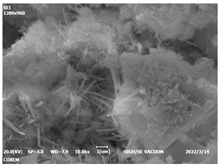	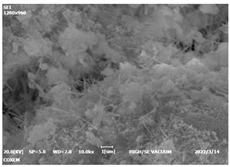
LC		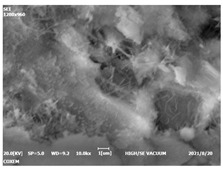	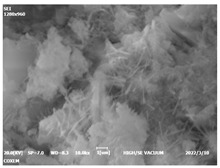	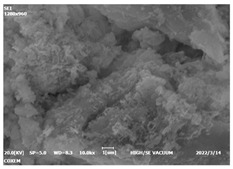
RC	3 d	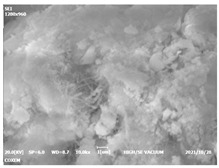	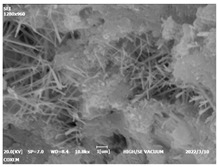	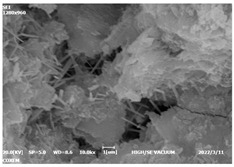
LC		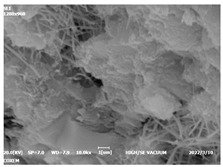	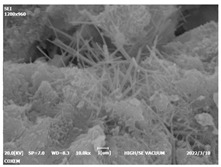	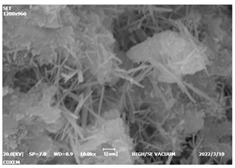

**Table 11 materials-16-04168-t011:** Degrees of hydration and morphology characteristics of hydration products at 12 h and 3 d.

Cement	Hydration Time		Blank	PC-1	PC-2
RC	12 h	Degree of hydration	0.2606	0.2278	0.0988
		Morphology	All flocculent	Subtle hydration product, no needle-like	Subtle hydration product, no needle-like
LC		Degree of hydration	0.2840	0.1193	0.0904
		Morphology	All flocculent	Subtle hydration product, no needle-like	Subtle hydration product, no needle-like
RC	1 d	Degree of hydration	0.5097	0.4950	0.4266
		Morphology	Needle-like morphology	Needle-like, length 1~2 μm	Few needle-shaped, partially fibrous products
LC		Degree of hydration	0.5147	0.4355	0.4125
		Morphology	Needle-like morphology	Needle-like, length 1 μm	A large quantity of fibrous products
RC	3 d	Degree of hydration	0.7304	0.7264	0.7180
		Morphology	More pores and needle-shaped crystals	Rod-shaped	Needle-shaped
LC		Degree of hydration	0.7510	0.6975	0.6595
		Morphology	More pores and needle-shaped crystals	Rod-shaped, needle-shaped	Rod-shaped, needle-shaped

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
