# Peer review of "Study on Dispersion, Adsorption, and Hydration Effects of Polycarboxylate Superplasticizers with Different Side Chain Structures in Reference Cement and Belite Cement"

_materials, 2023, doi:10.3390/ma16114168_

Round 1

Reviewer 1 Report

1)      “The PCE samples were characterized and tested using particle 14 charge analyzer, gel permeation chromatography, rheometer, total organic carbon analyzer, and 15 scanning electron microscopy.” Specify the type of rheological analysis (rotational rheometry?)

2)      “…and improve the plastic viscosity and rheological properties of the slurry.” Generic phrase, specify what would be an improvement! Reductions in plastic viscosity by how many percent?

3)      “The C-S-H gel generated by hydration of CC and PC-2 precipitated on the particle surface, inhibiting the acceleration of hydration, as evidenced by fewer fluffy substances observed in SEM.” This type of verification can only be made with a large volume of images that allows a statement of this nature.

4)      “Kong Xiangming et al. found that the higher the carboxyl group content in PCE…” Insert reference.

5)      “and the research results showed that the lower the side chain density, the better the flowability, lower yield stress, and lower plastic viscosity 58 of the cement paste [9].” Is this in relation to the initial dispersion? Make it clear in the sentence.

6)      “The chemical composition and particle size distribution of the cement is shown in Table 1 and Table 2, respectively.” Specify test conditions and equipment used.

7)      Present Table 3 after its citation in the text.

8)      “The chemical structures of PC-1 and PC-185 2 are shown in Figure 2a,b.” Explain the difference between the two superplasticizers.

9)      In item 2.3 insert a table with the detailed composition of the evaluated slurry.

10)   Specify the geometry and gap used in the rheometry test.

11)   “A water-to-cement ratio of 0.35 was used, and 100g of cement and a diluted admixture solution were weighed and thoroughly mixed.” Why was the w/c ratio varied compared to the other assays?

12)   The hydration stoppage was applied in the preparation of samples from which tests? This needs to be clearer.

13)   Mention scanning electron microscopy equipment and test conditions.

14)   “…and the dispersibility of PCE in CC is 294 also better.” This tendency needs to be justified. What characteristics of CC composition influence this behavior?

15)   “Under the same water reducer condition, the shear stress 323 and apparent viscosity of the standard cement are both higher than those of the CC, indicating that the standard cement slurry is thicker than the CC slurry” What characteristics of standard cement composition and physical properties influence this behavior?

16)   Has the hysteresis area of ​​the flow curves been evaluated? It can indicate system flocculation.

17)   Insert a figure with an example of the adjustment with the three mentioned.

18)   “From the data fitted by the Bingham model, it can be seen that under the same water reducer condition, the CC can significantly reduce the plastic viscosity of the cement slurry compared to the standard cement, and under the same cement condition, PC-1 can significantly reduce the plastic viscosity of the cement slurry compared to PC-2. This indicates that the use of CC and PC-1 can greatly improve the rheological properties of the cement slurry, which has a good correlation with the flowability results and follows the same trend.” The literature indicates that flowability normally correlates well with yield stress and not with viscosity. Reassess sentence.

19)   “This is mainly because the 347 charge density of PC-1 is higher than that of PC-2, and PC-1 has a larger Mark-Houwink a and a better comb-like structure compared to PC-2, with a smaller side chain molecular weight and molecular volume, ensuring that the negative charge in the main chain of PC-1 is not affected by the steric hindrance of the side chains during adsorption on the surface of silicate [42, 43], making PC-1 more capable of adsorbing on the surface of cement particles [44, 45], resulting in a higher adsorption amount and a good correlation with the dispersibility test results.” Very long and confusing sentence.

20)   Sea shell cement? Standardize nomenclature.

21)   How was the degree of hydration determined? Is it possible to reach this kind of conclusion with a few images? Review the entire discussion of micrographs.

Review all text. Extremely long sentences and difficult to understand.

Reviewer 2 Report

Review_2410678

Title

Study on Dispersion, Adsorption, and Hydration Effects of Polycarboxylate Superplasticizers with Different Side Chain Structures in Cement

Authors

Yunhui Fang , Xiaofang Zhang , Dongming Yan , Zhijun Lin , Xiuxing Ma , Junying Lai * , Yi Liu , Yuliang Ke , Zhanhua Chen , Zhaopeng Wang

Brief Summary

The use of PCE in construction is popular and in demand. Therefore, a comprehensive study of the effect of these types of plasticizers on cement systems is required.

Comments on the general concept

Statement of the problem.

The authors of the article set the goal of the study to study the effect of PCE not only on the rheological characteristics of cement pastes, but also on hydration processes, and also study the effect of a plasticizer on the morphology of the resulting crystalline hydrates.

The "Materials and Methods" section describes in sufficient detail the preparation of samples, and their investigation by means of modern physico-chemical methods of analysis. The results of the study are presented clearly, contribute to the scientific field of study, and are fully and correctly formulated. The conclusions are supported by the analysis of the results and provide answers to the research questions. The used literature corresponds to the stated topic of the study.

Remarks.

Some drawings are of poor resolution or need to be scaled up.

The shortcomings noted are not of a fundamental nature. They do not affect the scientific content of the article.

Conclusion

The scientific article is quite informative for specialists and is of interest to researchers involved in the problem of using the production and use of PCE in cement systems.

The article can be recommended for publication.

Author Response

Dear Sir/Madam,

   Thank you very much for taking the time to review and acknowledge our manuscript despite your busy schedule. We greatly appreciate your valuable feedback. In response to your suggestions, we have enlarged and enhanced the resolution of the scanning electron microscopy images in the manuscript.

With best regards,

                                                      Sincerely yours,

                                                     Xiaofang Zhang

Reviewer 3 Report

Manufacturing innovative superplasticizers is critical because of the rapid increase in specific cements and mineral addition usage.

This manuscript presents a ‘Study on Dispersion, Adsorption, and Hydration Effects of Polycarboxylate Superplasticizers with Different Side Chain Structures in Cement’. The manuscript is divided into four main parts: an introduction which provides an adequate background on superplasticizers effects, an Experimental section presenting the materials and methods, a Results section and a conclusion.

Overall the article is clear and sound, and may be of interest to the readers of the journal. Minor comments should be addressed:

1) Title: Maybe the use of Conch (Belite) cement can be mentioned since it it an interesting novelty of the paper

2) Abstract: RC and CC are not defined before usage

3) l 87: first period should be initial period

4) l 124: explain superplasticizer specificities rather than giving PCE22 name

5) l 134 please add a reference about the hydration kinetic model

6) Conch cement term is not very common worldwide. Can belite cement be used also?

7) Section 2.3.1 and 2.3.2: since Chinese standard may not be easily accessed, is it possible to add photographs of the flowability test and the rheometer configuration?

8) l 282: folded solid admixtures were used. Did this impair the mixing methodology?

9) Table 5: maybe it is better to place R² after the fitted parameters for the 3 models

10) Fig 4: indicate SPdosage in the caption

11) Section 3.4: the method for calculating t0, t1, … should be given in the Methods section

12) Table 7: Is it possible to calculate infinite heat (using 1/sqrt(time) graph)? Are these values the same for all the mixes?

13) Fig 8: increase fig size

14) l 676: Be careful Franck is the first name, same for Stefan and Joachim

Round 2

Reviewer 1 Report

Accept

Accept